# Raw Milk-Induced Protection against Food Allergic Symptoms in Mice Is Accompanied by Shifts in Microbial Community Structure

**DOI:** 10.3390/ijms22073417

**Published:** 2021-03-26

**Authors:** Suzanne Abbring, Phillip A. Engen, Ankur Naqib, Stefan J. Green, Johan Garssen, Ali Keshavarzian, Betty C. A. M. van Esch

**Affiliations:** 1Division of Pharmacology, Faculty of Science, Utrecht Institute for Pharmaceutical Sciences, Utrecht University, 3584 CG Utrecht, The Netherlands; suzanne.abbring@wur.nl (S.A.); j.garssen@uu.nl (J.G.); ali_keshavarzian@rush.edu (A.K.); 2Rush Center for Integrated Microbiome and Chronobiology Research, Rush Medical College, Rush University Medical Center, Chicago, IL 60612, USA; phillip_engen@rush.edu (P.A.E.); ankur_naqib@rush.edu (A.N.); 3Genomics and Microbiome Core Facility, Rush University Medical Center, Chicago, IL 60612, USA; stefan_green@rush.edu; 4Danone Nutricia Research, 3584 CT Utrecht, The Netherlands; 5Departments of Medicine and Physiology, Rush University Medical Center, Chicago, IL 60612, USA

**Keywords:** alkaline phosphatase, allergic diseases, food allergy, microbiota, milk processing, raw cow’s milk

## Abstract

The mechanism underlying the allergy-protective effects of raw cow’s milk is still unknown, but the modulation of the gut microbiome may play a role. The effects of consuming raw cow’s milk or processed milk on fecal microbial communities were therefore characterized in an experimental murine model. C3H/HeOuJ mice were treated with raw milk, pasteurized milk, skimmed raw milk, pasteurized milk supplemented with alkaline phosphatase (ALP), or phosphate-buffered saline (PBS) for eight days prior to sensitization and challenge with ovalbumin (OVA). Fecal samples were collected after milk exposure and after OVA sensitization, and microbiomes were characterized using 16S ribosomal RNA gene amplicon sequencing. Treatment with raw milk prior to OVA sensitization increased the relative abundance of putative butyrate-producing bacteria from the taxa *Lachnospiraceae* UCG-001, *Lachnospiraceae* UCG-008, and *Ruminiclostridium* 5 (Clostridial clusters XIVa and IV), while it decreased the relative abundance of Proteobacterial genera such as *Parasutterella*, a putative pro-inflammatory bacterial genus. This effect was observed after eight days of raw milk exposure and became more pronounced five weeks later, after allergic sensitization in the absence of milk. Similar trends were observed after treatment with skimmed raw milk. Conversely, the feeding of pasteurized milk led to a loss of allergy protection and a putative dysbiotic microbiome. The addition of ALP to pasteurized milk restored the protective effect observed with raw milk and mitigated some of the microbial community alterations associated with milk pasteurization. Raw milk-induced protection against food allergic symptoms in mice is accompanied by an increased relative abundance of putative butyrate-producing Clostridiales and a decreased relative abundance of putative pro-inflammatory Proteobacteria. Given the safety concerns regarding raw milk consumption, this knowledge is key for the development of new, microbiologically safe, preventive strategies to reduce the incidence of allergic diseases.

## 1. Introduction

The prevalence of allergic diseases such as allergic rhinitis, atopic dermatitis, asthma, and food allergy has increased dramatically in recent decades and has become a major health problem, particularly in affluent and fast-developing societies [1]. Due to the severe impact on the patient’s quality of life and extensive healthcare costs, this vast prevalence has major socio-economic consequences [2]. Even though therapies for allergic diseases have improved over the years with the introduction of allergen-specific immunotherapy and the development of new pharmacological agents aimed at combating inflammatory responses, as well as providing symptomatic relief, these therapies have remained largely noncurative [3,4,5].

Changes in environmental exposures and lifestyle practices such as diet, microbiome, hygiene, pollutants, exercise, and infections are believed to be associated with the increase in and severity of allergic diseases [1]. Numerous studies have emphasized the interaction of such factors with the immune system leading to modified immune responses [6,7,8]. A well-known example is the “farm effect” wherein children growing up on a farm have a lower risk of developing allergic diseases compared to children who are raised in the same rural area but not on a farm [9,10,11,12,13]. This beneficial “farm effect” has been mainly attributed to the consumption of raw, unprocessed cow’s milk [14,15,16,17,18,19]. Since the allergy-protective raw milk effect also has been observed in children not exposed to a farming environment [14,15,16], raw cow’s milk consumption could provide a potential natural solution for allergic diseases.

In previous work, we strengthened the epidemiological findings for an asthma- and allergy-protective effect of raw cow’s milk consumption by demonstrating a causal relationship using preclinical murine models [20,21]. In addition, we showed that the heat-sensitive components, not the fat content, are responsible for the raw milk-induced protection against food allergic symptoms [22]. These studies have increased our knowledge about the protective raw milk constituents, although the bioactive component(s) remain to be elucidated.

Current knowledge on the mechanism(s) underlying the allergy-protective effects of raw cow’s milk is still limited. It is hypothesized that the many immunomodulatory components present in raw milk create a regulatory environment favoring unresponsiveness upon allergen exposure [23]. Potential mechanisms of action are improved intestinal barrier function, the promotion of regulatory T-cell development, and modulation of the gut microbiome [23,24,25]. 

The gut microbial composition and functionality is influenced by dietary elements and can in turn influence the host immune response with effects on asthma and allergy development [26]. Interestingly, several components present in raw cow’s milk may modulate the gut microbiome. Proteins such as lactoferrin, lactoperoxidase, and lysozyme have antimicrobial activity, and milk oligosaccharides such as sialyllactose can promote the growth of probiotic Bifidobacteria that in turn produce short-chain fatty acids (SCFA), known to prevent the development of allergic diseases [23,26,27]. In addition, raw cow’s milk has been shown to contain significantly higher abundances of viable microbial cells compared to heat-treated milk [18]. Nevertheless, compelling evidence for a raw milk-induced change in the gut microbiome is still lacking. 

In the current study, we therefore analyzed the gut microbiota of mice treated with raw milk and then sensitized and challenged with ovalbumin (OVA) to induce food allergy. These mice were previously shown to be protected against food allergic symptoms by raw milk [22]. We also previously demonstrated that the suppression of food allergic symptoms by raw milk was retained after skimming but eliminated after pasteurization of the milk. Supplementing pasteurized milk with alkaline phosphatase (ALP), one of the first bioactive raw milk components that loses its activity upon heat treatment, restored the protective effect [22]. In the current study, we compared fecal microbial communities of mice undergoing different milk treatments to assess the effects on microbial community structure and to assess whether changes in the gut microbiota contribute to observed allergy-protective effects.

## 2. Results

### 2.1. Microbiome Sequencing

Fecal samples (*n* = 100: day −9 (*n* = 8); day −1 (phosphate-buffered saline (PBS) *n* = 14, raw, pasteurized, skimmed, pasteurized + ALP *n* = 8 per group); and day 31 (PBS *n* = 6, OVA, raw, pasteurized, skimmed, pasteurized + ALP *n* = 8 per group); Figure 1) were processed for microbiome sequencing. A total of 4,099,786 sequences were generated, with an average depth of sequencing of 40,998 per sample (median = 41,629; min = 7061; max = 63,657). Sequence data were rarefied to a depth of 7000 sequences per sample for diversity analyses (Appendix A). At the start of the study (day −9), before tolerance induction, the mouse fecal samples (*n* = 8) microbial communities were similar, with a Bray–Curtis within-group similarity index average of 73.48 ± 4.318 SD, and baseline samples had Bray–Curtis similarity values greater than 70%.

### 2.2. Microbial Communities Are Highly Similar between Treatments Directly after Milk Exposure

Fecal samples were collected after the tolerance induction period, directly after PBS/milk treatment (day −1; Figure 1) and assessed for differences in microbial community structure. Microbial communities were broadly similar at the day −1 sampling and were dominated by bacteria from the phyla Firmicutes and Bacteroidetes, and bacteria from the genera *Muribaculaceae*, *Lactobacillus*, *Lachnospiraceae*, *Bacteroides*, and *Roseburia* (>75% of all sequences at day −1; Figure 2A,B). With the exception of genus-level richness, which was increased in pasteurized milk-treated mice relative to raw milk-treated mice (one-way ANOVA: F(_4,41_) = 2.702, *p* = 0.0436; Tukey post hoc: *p* = 0.028), no significant differences in alpha diversity metrics (i.e., Shannon index, Simpson’s index, richness, and evenness) between groups were observed (Appendix A).

In addition, beta diversity analyses (i.e., analysis of similarity (ANOSIM)) were performed to determine if bulk microbial communities were significantly different. At the day −1 sampling, the observed microbial community structures of most treatment groups were not significantly different (ANOSIM R-values from −0.098 to 0.16; *p*-values from 0.057 to 0.897). However, the fecal microbial community structure of the raw milk treatment group was significantly different from the pasteurized milk group (R = 0.286, *p* = 0.003) and from the skimmed raw milk group (R = 0.288, *p* = 0.010; Table 1).

Features analyses were conducted at the taxonomic levels of phylum, family, and genus to identify microbial taxa that were significantly differently abundant between treatment groups at the day −1 sampling. Kruskal–Wallis tests were performed on centered log ratio transformed (CLR) relative abundance data, and false discovery rate (FDR)-corrected *p*-values and Dunn’s multiple group comparison’s post hoc test adjusted *p*-values are reported (Appendix A). No significant differences (FDR *p*-values < 0.05) in the relative abundance of individual taxa were observed between the treatment groups at the day −1 sampling (Appendix A). 

We further assessed the relative abundance of putative butyrate-producing taxa together, including multiple *Lachnospiraceae, Roseburia*, and *Blautia* genera. No significant differences in the combined relative abundance of these taxa between treatment groups were observed at the day −1 sampling (Appendix A).

### 2.3. Significant Differences in Microbial Community Structure Primarily Manifested Themselves after Sensitization

At day 31, after the last sensitization, mouse fecal microbiomes were characterized again. Microbial communities were broadly similar again at the day 31 sampling and dominated by bacteria from the phyla Firmicutes and Bacteroidetes, and bacteria from the taxa *Muribaculaceae*, *Lactobacillus*, *Lachnospiraceae*, *Bacteroides*, and *Roseburia* (>75% of all sequences at day 31; Figure 2C,D). However, across all groups, the relative abundance of Bacteroidetes increased relative to day −1 (Figure 2A,C), as did the relative abundances of genera *Muribaculaceae* and *Lactobacillus* (Figure 2B,D). At the day 31 sampling, no significant differences in fecal microbiome Shannon index, Simpson’s index, and evenness between groups were observed (Appendix A). However, genus-level richness values varied significantly between groups, with OVA-sensitized allergic control mice (OVA group) and pasteurized milk-treated mice having significantly higher richness relative to PBS-sensitized control mice (PBS group) and raw milk-, skimmed raw milk-, and pasteurized milk + ALP-treated mice (Figure 3; one-way ANOVA: F(_4,41_) = 15.19, *p* ˂ 0.0001).

Beta-diversity analyses (i.e., ANOSIM) demonstrated that at the day 31 sampling, the fecal microbial communities of raw milk- and skimmed raw milk-treated mice were no longer significantly different (R = 0.122, *p* = 0.106), while the differences between raw milk and pasteurized milk persisted (R = 0.226, *p* = 0.013; Table 1). Microbial community structure in all other treatment groups were significantly different from each other (Table 1; R-values from 0.159 to 0.610; *p*-values from 0.002 to 0.049).

At day 31, nine microbial features were trending as different (FDR-*p* ≤ 0.07) or significantly different (FDR-*p* < 0.05) between groups, including the phylum Proteobacteria, family Burkholderiaceae (Proteobacteria), genus *Oscillibacter* (Firmicutes), genus *Ruminiclostridium* (Firmicutes), genus *Parasutterella* (Proteobacteria), and multiple Lachnospiraceae genera (*Blautia*, NK4A136 group, UCG-001, and UCG-008; Firmicutes; Table 2). The relative abundances of Gram-negative, putative pro-inflammatory phylum Proteobacteria (i.e., Burkholderiaceae and *Parasutterella*) were significantly higher in pasteurized milk-treated mice and pasteurized milk + ALP-treated mice compared to PBS-treated allergic control mice (OVA group) and raw milk-treated mice (Table 2). The relative abundance of bacteria from the *Lachnospiraceae* NK4A136 group (putative butyrate producers) was significantly increased in skimmed raw milk-treated mice compared to PBS-treated allergic control mice (OVA group; Table 2).

The relative abundances of *Blautia*, *Oscillibacter*, and *Lachnospiraceae* UCG-008 were increased in OVA-sensitized allergic control mice compared to PBS-sensitized control mice (Table 2). Treating mice with raw milk or skimmed raw milk prior to OVA sensitization significantly increased the relative abundances of *Lachnospiraceae* UCG-001, *Lachnospiraceae* UCG-008, and *Ruminiclostridium* 5 compared to PBS-treated allergic control mice (OVA group; Table 2). Bacteria from the *Lachnospiraceae* UCG-001 group were elevated in most milk treatments, but not in the pasteurized milk treatment (Table 2). Of the nine microbial trending and significant features, *Lachnospiraceae* UCG-001 is the only feature significantly different by Dunn’s post hoc test between mice in the pasteurized milk treatment and mice in the pasteurized milk + ALP treatment (Table 2).

The percent relative abundance of putative butyrate-producing genera (Kruskal–Wallis: *p* = 0.0055), and the ratio of total butyrate-to-total SCFA (Kruskal–Wallis: *p* = 0.0012) were significantly different across treatment groups at day 31 (Figure 4A,B). Specifically, skimmed raw milk-treated mice had higher relative abundances of putative butyrate producers when compared to PBS-treated allergic control mice (Butyrate (Dunns: *p* = 0.020); Ratio (Dunns: *p* = 0.013); Figure 4A,B).

A machine-learning algorithm (Boruta) was used to further identify taxa differentiating microbial communities by treatment at the day 31 sampling. This analysis identified the genera *Parasutterella* and *Bifidobacterium* were particularly driving the differences between groups observed at day 31 (Appendix A). *Parasutterella* were identified by using a taxon-by-taxon Kruskal–Wallis test (above), while *Bifidobacterium* were not significant (FDR-*p* = 0.162; Appendix A) by this analysis. We note, however, that *Bifidobacterium* levels were extremely low in all treatment groups except for OVA (PBS-treated allergic mice) and pasteurized milk treatments (Appendix A).

## 3. Discussion

We previously showed that raw, unprocessed cow’s milk suppressed allergic symptoms in a murine model for food allergy [22]. This protective effect was retained after skimming, but lost upon pasteurization of the milk, indicating that heat-sensitive components could underlie the allergy-protective effects of raw cow’s milk. Additionally, we showed that ALP, a heat-sensitive bioactive raw milk component, was able to partially restore the allergy-protective effect lost with heat treatment. The observed protection against food allergic symptoms was accompanied by a reduction in allergen-specific Th2 responsiveness, and an induction of tolerance-associated cell types, such as CD103^+^ dendritic cells (DCs) and regulatory T-cells (Tregs). Moreover, SCFA concentrations in the cecum hinted towards a potential immune modulation via the gut microbiome [22]. In the present study, we therefore interrogated mouse fecal samples to determine if changes in the microbial community structure are induced by raw milk treatment and to determine the consequences of milk processing steps on the fecal microbiome.

In this study, analysis of the fecal microbiome of mice indicated that raw milk exposure increased the relative abundance of several putative butyrate-producing bacterial taxa from the order Clostridiales, including *Lachnospiraceae* UCG-001, *Lachnospiraceae* UCG-008, and *Ruminiclostridium* 5, while it decreased the relative abundance of some Proteobacterial taxa (e.g., *Parasutterella*). This effect was visible after eight days of raw milk exposure (i.e., after the tolerance induction period) but became more pronounced five weeks later, after allergic sensitization in the absence of milk. Similar to animals treated with raw cow’s milk, animals treated with skimmed raw cow’s milk prior to OVA sensitization had higher relative abundances of putative anti-inflammatory bacteria and lower relative abundances of pro-inflammatory bacteria. Moreover, the relative abundance of all identified putative butyrate-producing genera was higher in raw milk and skimmed raw milk treatments relative to all other samples, and significantly higher in the skimmed raw milk treatment relative to PBS and OVA treatments. Mice treated with pasteurized milk had relatively high relative abundances of putative pathobiont inflammatory bacteria and low relative abundances of putative anti-inflammatory bacteria, and this may represent a dysbiotic microbiome. The addition of ALP to pasteurized milk increased the relative abundances of anti-inflammatory bacteria, but did not reduce pro-inflammatory bacteria levels. Remarkably, allergic sensitization to OVA increased bacterial richness compared to sham sensitization. This finding was not observed in any of the other groups, except in the group treated with pasteurized milk prior to OVA sensitization. 

Raw cow’s milk consumption has been reported to protect against childhood asthma and allergies [16,17,18,19,20]. If the risk of infections could be overcome, consumption of raw milk would be an attractive preventive strategy for allergic diseases. Further research into the underlying mechanisms is therefore highly desirable. One of the hypothesized working mechanisms of raw cow’s milk is modulation of the gut microbiome and its metabolic activity (i.e., SCFA production) [23,24,25]. Recently, an observational study was conducted in which the impact of unpasteurized dairy products on the human gut microbiome was investigated in participants undertaking a residential 12-week cookery course on an organic farm [28]. This study was one of the first to show that the intake of unpasteurized milk and dairy products increased the relative abundance of putative probiotic bacteria [28]. However, due to the observational nature of this study, authors were unable to disentangle the impact of diet and geographical environment on the gut microbiome.

In our study, fecal samples were collected after milk exposure and after sensitization to OVA. We investigated whether the different milk types induced changes in the microbiota composition, and whether these changes could be associated with the allergy-protective effects previously observed after raw milk, skimmed raw milk, and pasteurized milk + ALP exposure [22]. One of the most striking findings was that raw and skimmed raw milk exposure increased the relative abundance of genera belonging to the butyrate-producing Clostridiales clusters XIVa and IV (informal nomenclature: *Clostridium* clusters), including the taxa *Lachnospiraceae* UCG-001, *Lachnospiraceae* UCG-008, and *Ruminiclostridium* 5 [29,30,31]. Both raw milk types showed the strongest increase in relative abundance of these bacteria at day 31 (after sensitization), five weeks after the last milk exposure. This could indicate that bacteria present in raw milk needed time to colonize in the gut or that raw milk components promoting the growth of probiotic bacteria, such as sialyllactose [23,32] and lactoferrin [33,34,35], needed time to exert their effect. 

Bacteria within the *Clostridium* clusters XIVa and IV are known for their high level of metabolite production. They can produce acetic, propionic, and butyric acids through the degradation of non-digestible oligosaccharides. The organisms typically reside in close proximity to the host epithelium, and they can exert a strong influence on the host immune system [36,37]. For example, the colonization of mice with a defined mixture of Clostridial strains belonging to clusters XIVa and IV induced the accumulation and differentiation of colonic FoxP3^+^ Treg cells [38]. The presence of *Clostridium* induced the release of active TGF-β and other Treg-inducing factors from colonic epithelial cells and thereby presumably conditioned CD103^+^ DCs to drive Treg differentiation. Furthermore, *Clostridium* species have also been shown to induce IL-10-expressing Tregs in the colon [38]. Importantly, the oral inoculation of a mixture of these species resulted in resistance to allergy, as demonstrated by reduced Th2 responsiveness in the spleen and lower systemic IgE responses [38].

Although we did not assess Treg numbers in the colon, we did observe an increase in TGF-β- and IL-10-producing Treg cells in lymphoid organs after raw milk treatment [22,39]. In addition, raw milk exposure induced tolerogenic CD103^+^ DCs [22]. Since it is well known that Treg cells travel between compartments [40] and that bacterial metabolites such as SCFAs can enter the systemic circulation [41], we speculate that the induction of SCFA-producing Clostridiales by raw milk underlies the observed increase in tolerance-associated cell types in lymphoid organs which may have contributed to the observed allergy-protective effect. This hypothesis is supported by the estimation of the total relative abundance of all identified putative butyrate-producing taxa in mouse feces. These measurements indicated a significantly higher relative abundance of total butyrate-producing genera in skimmed raw milk-treated mice compared to PBS-treated allergic control mice at the day 31 sampling. For raw milk-treated mice, a similar pattern was observed although the difference was not significant at the *p* < 0.05 level (*p* = 0.233). Interestingly, bacterial communities with the potential of producing butyrate were also found to be associated with the protective farm effect on childhood asthma in the PASTURE birth cohort study [42]. 

The current literature particularly speculates about a raw milk-induced outgrowth of Bifidobacteria and Lactobacilli [23,24,25]. In line with this hypothesis, the previously mentioned observational study demonstrated an increase in the relative abundance of Lactobacilli following the consumption of unpasteurized milk and dairy products [28]. However, in our study, the relative abundance of both probiotic bacterial genera was low in raw milk-treated mice. Our findings are consistent with another study which showed that the relative abundance of *Bifidobacterium* members was lower in the feces of infants raised in farming versus nonfarming environments [43]. Solid evidence is therefore still lacking on the role of these putative probiotic bacterial taxa in allergic responses. Strikingly, in our study, the relative abundances of Bifidobacteria and Lactobacilli were highest in PBS-treated allergic control mice and pasteurized milk-treated mice, while these mice showed the strongest allergic symptoms. 

Another remarkable observation in this study is of the higher microbial richness in PBS-treated allergic control mice (i.e., OVA group) and pasteurized milk-treated mice when compared to all other groups at the day 31 sampling. A high microbial diversity is usually linked to improved gut health [44], and several studies have found an association between low microbial diversity and allergic diseases such as eczema and allergic rhinitis [45]. Elsewhere, bacterial diversity was found to be increased in sputum samples of asthmatic patients, compared to non-asthmatic subjects [46,47]. These results do not suggest a straightforward relationship between fecal microbial diversity and allergic responses.

Besides increasing the relative abundance of anti-inflammatory bacteria, raw milk treatment also decreased the relative abundance of pro-inflammatory bacteria. Mice in the raw milk treatments had both higher relative abundance of putative anti-inflammatory bacteria and lower relative abundance of putative pro-inflammatory bacteria, especially relative to animals in the pasteurized milk treatment. In particular, Proteobacteria levels were greatly reduced in raw milk treatments, relative to levels in pasteurized milk-treated mice. Proteobacteria are generally recognized as the microbial signature of gut dysbiosis, a condition that has been characterized by a microbial imbalance leading to metabolic disorders and intestinal inflammation [48]. In this study, levels of bacteria from the putative pathobiont inflammatory bacterial genus *Parasutterella* were most strongly associated with pasteurization. *Parasutterella* levels have been previously associated with food allergy in infancy [49]. Although we did not observe an increase in the relative abundance of *Parasutterella* in OVA-sensitized allergic control mice compared to PBS-sensitized control mice, the relative abundance of *Parasutterella* did increase significantly in pasteurized milk-treated mice compared to raw milk-treated mice.

The addition of ALP to pasteurized milk was previously shown to partially restore the allergy-protective effect lost upon heat treatment [22]. Interestingly, the fecal microbiota community structure of pasteurized milk + ALP-treated mice also differed significantly from mice treated with pasteurized milk alone. By adding ALP to pasteurized milk, the relative abundance of anti-inflammatory bacteria (e.g., *Lachnospiraceae* UCG-001) increased, and even reached similar bacterial levels as observed in raw milk-treated mice. Spiking pasteurized milk with ALP did not, however, lower the relative abundance of Proteobacteria. Since mice treated with pasteurized milk + ALP were still protected against food allergic symptoms, these results suggest that inducing butyrate-producing, anti-inflammatory bacteria is more important than reducing pro-inflammatory bacteria levels in preventing allergic diseases.

In summary, we demonstrated that the raw milk-induced protection against food allergic symptoms coincides with an increased relative abundance of putative butyrate-producing genera and a decreased relative abundance of Proteobacterial taxa in the feces of mice long after raw milk exposure. Similar microbial responses to raw milk and skimmed raw milk were observed. The previously observed loss of allergy protection after pasteurization was accompanied by a putative dysbiosis-associated microbial phenotype that could be partly restored by the addition of ALP. The effect of the origin of the milk (e.g., organic vs. conventional) on the allergy-protective capacity and associated microbiota changes needs to be assessed in future studies.

By showing a raw milk-induced modulation of the gut microbiome, this study contributes to the current knowledge on mechanisms underlying the allergy-protective effects of raw cow’s milk. Since the consumption of raw cow’s milk is not recommended due to the potential presence of pathogens, this knowledge is crucial to promote the development of microbiologically safe alternatives.

## 4. Materials and Methods

### 4.1. Mice

Three-week-old, specific-pathogen-free, female C3H/HeOuJ mice (Charles River Laboratories, Sulzfeld, Germany) were housed at the animal facility of Utrecht University (Utrecht, The Netherlands) in filter-topped Makrolon cages (one cage/group, *n* = 6–8/cage) with standard chip bedding, Kleenex tissues, and a plastic shelter on a 12 h light/dark cycle with access to food (“Rat and Mouse Breeder and Grower Expanded”; Special Diet Services, Witham, UK) and water ad libitum. Upon arrival, mice were randomly assigned to the control or experimental groups and were acclimatized to the laboratory conditions for one week prior to experimentation. All animal procedures were approved by the Ethical Committee for Animal Research of the Utrecht University and complied with the European Directive 2010/63/EU on the protection of animals used for scientific purposes (AVD1008002015346).

### 4.2. Milk Types

Raw, unprocessed cow’s milk was collected from a dairy farm (Macroom, Ireland) and subsequently divided into three aliquots. Aliquot 1 was stored directly at −20 °C with no further treatment (raw milk). Aliquot 2 was heated at 78 °C for 15 s, cooled on ice to 4 °C and then stored at −20 °C (pasteurized milk) and aliquot 3 was skimmed using a centrifugal separator at 55 °C to remove milk fat and then cooled on ice to 4 °C and stored at −20 °C (skimmed milk; 0.1% fat). All milk types were solely produced for experimental purposes (Danone Nutricia Research, Macroom, Ireland). Before use, milk samples were thawed at room temperature and a portion of the pasteurized milk was supplemented with bovine intestinal ALP (pasteurized milk + ALP; 3 units/0.5 mL pasteurized milk; kindly provided by Prof. Dr. W. Seinen (Alloksys Life Sciences BV/AMRIF BV, Wageningen, The Netherlands)).

### 4.3. Experimental Design—Tolerance Induction, Sensitization and Challenges

The experimental design of the study is illustrated in Figure 1. Mice were orally treated by gavage once daily with 0.5 mL raw milk, pasteurized milk, skimmed raw milk, pasteurized milk supplemented with ALP, or PBS (as a control) for eight consecutive days (experimental days −9 to −2) after acclimatization. After this tolerance induction period, mice were orally sensitized with 20 mg of the chicken egg protein OVA (grade V; Sigma–Aldrich, Zwijndrecht, The Netherlands) dissolved in 0.5 mL of PBS containing 10 µg of cholera toxin (CT; List Biological Laboratories, Campbell, CA, USA) as an adjuvant (days 0, 7, 14, 21, and 28; *n* = 8/group). PBS-sensitized control mice (*n* = 6) received CT alone (10 µg/0.5 mL of PBS). On days −9, −1, and 31 (before tolerance induction, after tolerance induction, and after sensitization), fecal samples were collected for microbiota analysis. On day 33, five days after the last sensitization, all mice were intradermally and orally challenged with OVA (10 µg of OVA/20 µL of PBS and 50 mg of OVA/0.5 mL of PBS, respectively) to assess the allergic response (results previously published [22]). Sixteen hours after the oral challenge (day 34), mice were killed by cervical dislocation.

### 4.4. Microbiota Profiling and Bioinformatics Analysis

Total genomic DNA was extracted from mouse fecal samples utilizing the QIAamp DNA Microbiome Kit (QIAGEN, Germantown, MD, USA), and extract DNA concentrations were measured using fluorometric quantitation (Qubit, Life Technologies, Grand Island, NY, USA). Primers 515F/806R (515F: GTGYCAGCMGCCGCGGTAA; 806R: GGACTACNVGGGTWTCTAAT), targeting the V4 variable region of microbial small subunit (SSU or 16S) ribosomal RNA (rRNA) genes, were used for PCR (e.g., Caporaso et al. [50]), and prepared for high-throughput amplicon sequencing using a two-stage PCR method, as described previously [51]. Negative controls were used with each set of amplifications, which indicated no background contamination. Sequencing was performed using an Illumina MiSeq (Illumina, San Diego, CA, USA), with a V2 kit and paired-end 250 base reads at the University of Illinois at Chicago (UIC) Genome Research Core (GRC). Raw sequence data (FASTQ files) were deposited in the National Center for Biotechnology Information (NCBI) Sequence Read Archive (SRA) under the BioProject identifier PRJNA576858. 

Basic processing of the raw data was performed by the Research Informatics Core (RIC) at UIC. Raw FASTQ forward and reverse read files for each sample were merged using the software package PEAR (Paired-End-reAd mergeR; v0.9.8; (http://www.exelixis-lab.org/web/software/pear (accessed on 1 February 2021)) [52,53]. Sequences were then processed using the software package QIIME, v1.9 [54]. Briefly, merged reads were quality trimmed and primer sequences were removed. Sequences shorter than 225 base pairs were discarded. Chimeric sequences were identified and removed using the USEARCH algorithm [55], along with the annotation being performed against the SILVA (silva_132_16S.97 [56]) reference database at a similarity. A biological observation matrix (BIOM) [57] was generated at each taxonomic level from phylum to species. Downstream statistical analyses were performed using the software packages Primer7, as well as *vegan* (https://cran.r-project.org/package=vegan (accessed on 1 February 2021)) and *boruta* (http://www.jstatsoft.org/v36/i11/ (accessed on 1 February 2021)), both within the R programming environment [58,59,60,61].

### 4.5. Statistics

Analyses of alpha diversity and beta diversity were used to examine changes in fecal microbial community structure between animals and between experimental groups. Alpha diversity indices (i.e., Shannon, Simpson, richness, and Pielou’s evenness) were generated using the package “vegan” implemented in the R programming environment (https://cran.r-project.org (accessed on 1 February 2021), https://github.com/vegandevs/vegan accessed date: 30 December 2019). Alpha diversity indices were calculated on datasets rarefied to 7000 sequences per sample [62]. Differences in alpha diversity indices between groups were assessed for significance using one-way analysis of variance (ANOVA) tests with Tukey’s post hoc test adjusted *p*-values reported.

To examine differences in microbial community structure between samples, a pairwise Bray–Curtis dissimilarity (non-phylogenetic) metric was generated using the Primer7 software package (PRIMER-E Ltd., Lutton, United Kingdom), and analysis of similarity (ANOSIM) calculations were performed on pair-wise distance matrices to determine if differences in microbial community structure between treatments were significant. ANOSIM was performed at the taxonomic level of genus on square-root transformed data. Before tolerance induction (day −9), the fecal microbial community structure was examined using within-group Bray–Curtis similarity scores. Random forest (RF) models were used to predict allergy status based on the genus-level bacterial profiles using the default parameters of the Boruta algorithm (“randomForest” package within R; (http://www.jstatsoft.org/v36/i11/ (accessed on 1 February 2021)) [61]. Differences in the relative abundance of individual taxa across treatment groups were assessed for significance using the Kruskal–Wallis test on centered log ratio transformed (CLR) data, with false discovery rate (FDR)-corrected *p*-values, plus Dunn’s multiple group comparison’s post hoc test. A predictive model depicting the relative abundances of butyrate-producing taxa and total butyrate-to-total SCFA ratio was examined in the mice groups, as previously described [63,64]. Figures were created using GraphPad Prism (v8.00; GraphPad Software Inc, La Jolla, San Jose, CA, USA).

## Figures and Tables

**Figure 1 ijms-22-03417-f001:**
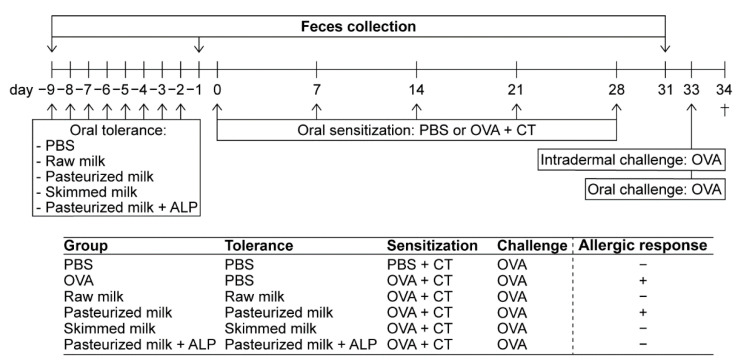
Schematic overview of the study design. Female C3H/HeOuJ mice were grouped as depicted: PBS (PBS-sensitized control mice; *n* = 6), OVA (OVA-sensitized allergic mice; *n* = 8), raw milk (raw milk-treated mice; *n* = 8), pasteurized milk (pasteurized milk-treated mice; *n* = 8), skimmed milk (skimmed raw milk-treated mice; *n* = 8), and pasteurized milk + ALP (pasteurized milk + ALP-treated mice; *n* = 8). On experimental days 0, 7, 14, 21, and 28, mice were orally sensitized to the hen’s egg protein OVA using cholera toxin as an adjuvant (20 mg of OVA + 10 µg of CT/0.5 mL of PBS). PBS-sensitized control mice received cholera toxin alone. Prior to sensitization, mice were orally treated with 0.5 mL of PBS (as a control), raw milk, pasteurized milk, skimmed raw milk, or pasteurized milk supplemented with ALP for eight consecutive days (day −9 to −2). On days −9, −1, and 31 (before tolerance induction, after tolerance induction, and after sensitization), fecal samples were collected for microbiota analysis. On day 33, all mice were intradermally (10 µg of OVA/20 µL of PBS) and orally (50 mg of OVA/0.5 mL of PBS) challenged with OVA. Mice were killed on day 34 (as indicated by †). The allergic response in each group is displayed with a “+” for an allergic response or a “–” for no allergic response (results previously published [22]). PBS, phosphate-buffered saline; ALP, alkaline phosphatase; OVA, ovalbumin; CT, cholera toxin.

**Figure 2 ijms-22-03417-f002:**
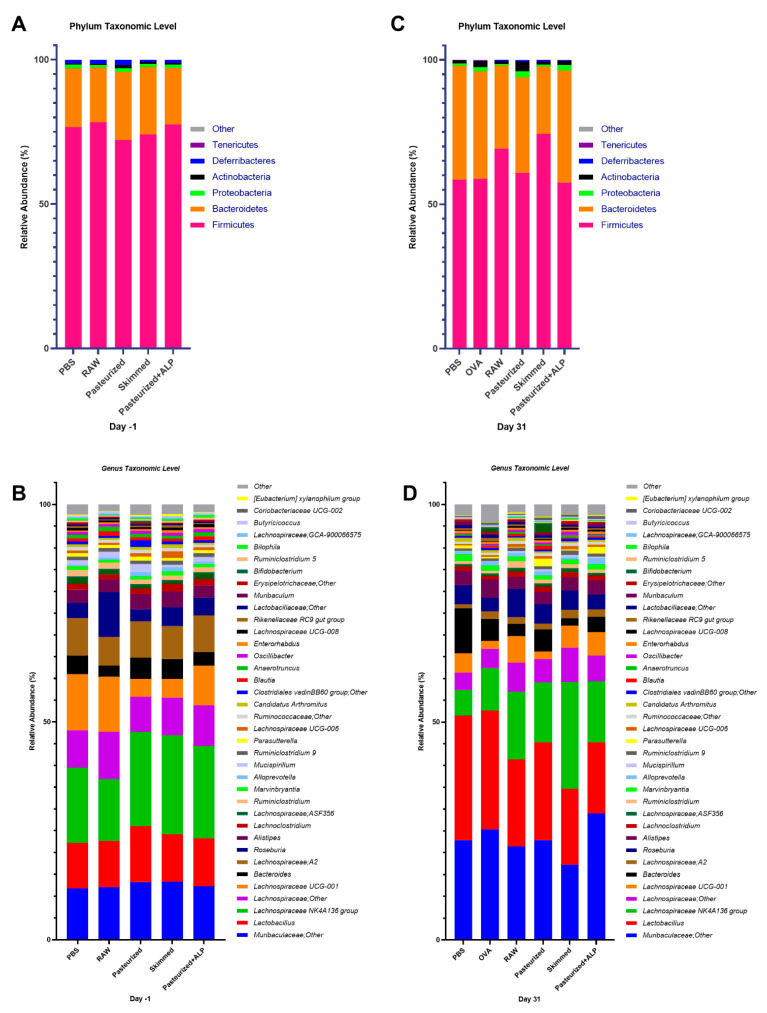
Microbial profiles of the control and experimental groups at day −1 and day 31. (**A**) Day −1 stacked column plots depicting the percent mean relative abundance (>1%) of bacterial phyla. (**B**) Day −1 stacked column plots depicting the percent mean relative abundance (>0.1%) of bacterial genera. (**C**) Day 31 stacked column plots depicting the percent mean relative abundance (>1%) of bacterial phyla. (**D**) Day 31 stacked column plots depicting the percent mean relative abundance (>0.1%) of bacterial genera. PBS, phosphate-buffered saline; OVA, ovalbumin; raw, raw cow’s milk; pasteurized, pasteurized cow’s milk; skimmed, skimmed raw cow’s milk; pasteurized + ALP, pasteurized milk spiked with alkaline phosphatase.

**Figure 3 ijms-22-03417-f003:**
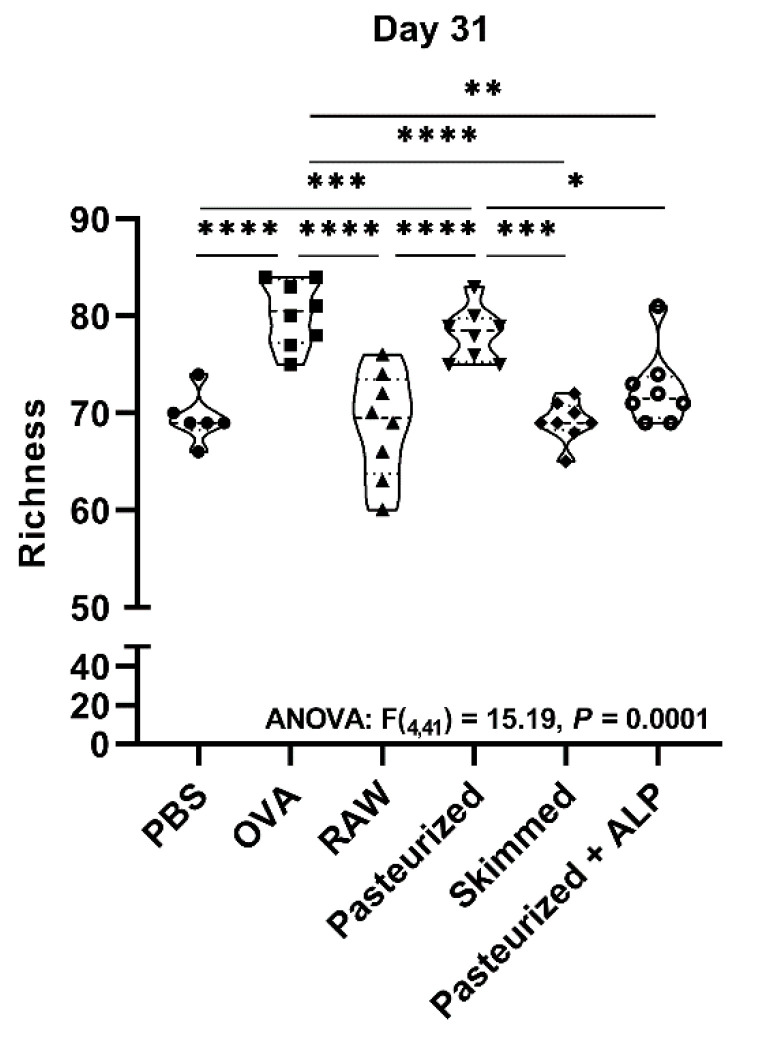
Ovalbumin sensitization and pasteurized milk treatment significantly increased bacterial richness at day 31. Microbial richness was examined at the taxonomic level of genus. Measurements were based on 16S rRNA gene amplicon sequence datasets rarefied to 7000 sequences per sample. One-way ANOVA results and Tukey post hoc test adjusted *p*-values shown: * *p* < 0.05, ** *p* < 0.01, *** *p* < 0.001, **** *p* < 0.0001. PBS, phosphate-buffered saline; OVA, ovalbumin; raw, raw cow’s milk; pasteurized, pasteurized cow’s milk; skimmed, skimmed raw cow’s milk; pasteurized + ALP, pasteurized milk spiked with alkaline phosphatase.

**Figure 4 ijms-22-03417-f004:**
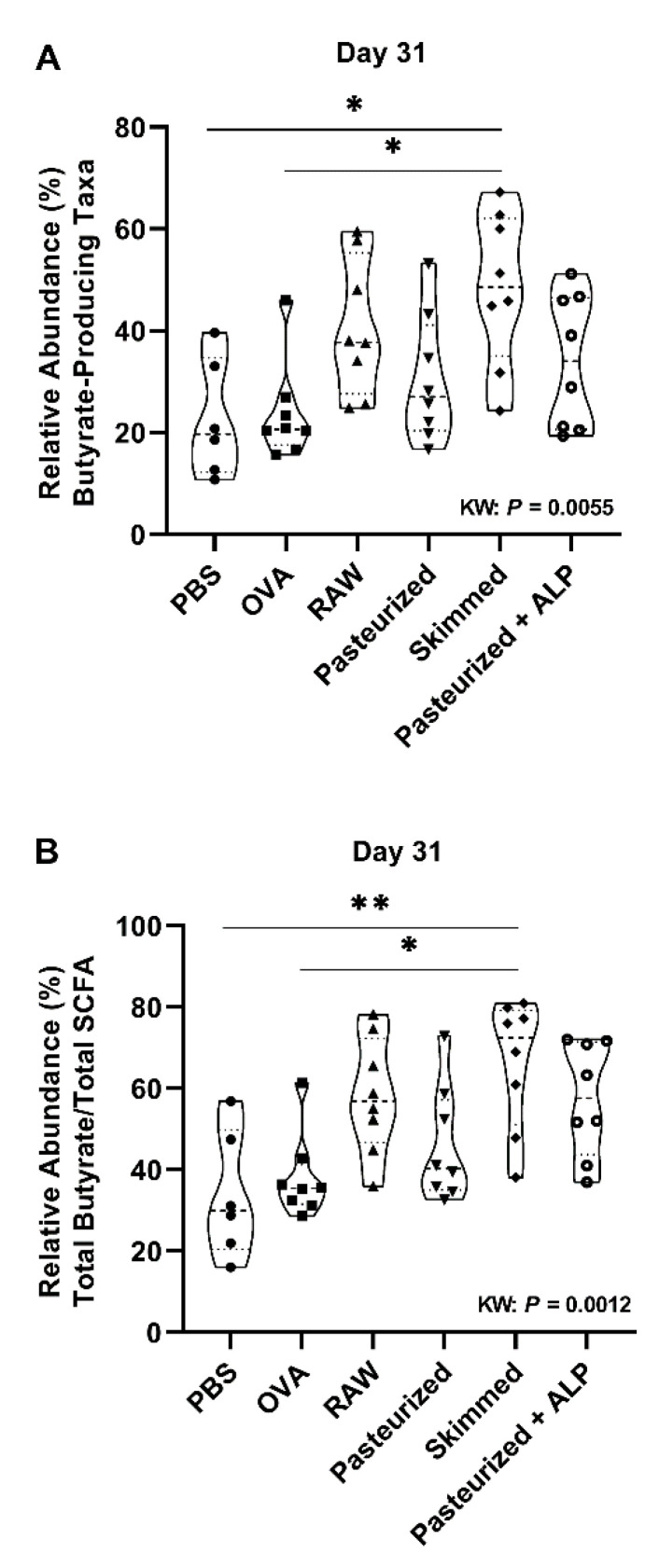
Skimmed raw milk is associated with increased butyrate-producing genera at day 31. The percent mean relative abundance of (**A**) putative butyrate-producing taxa and (**B**) total butyrate-to-total SCFA ratio examined is shown for the different control and experimental groups at day 31. Kruskal–Wallis results and Dunn’s post hoc test adjusted *p*-values shown: * *p* < 0.05, ** *p* < 0.01. PBS, phosphate-buffered saline; OVA, ovalbumin; raw, raw cow’s milk; pasteurized, pasteurized cow’s milk; skimmed, skimmed raw cow’s milk; pasteurized + ALP, pasteurized milk spiked with alkaline phosphatase; KW, Kruskal–Wallis; SCFA, short-chain fatty acids.

**Table 1 ijms-22-03417-t001:** Group analysis of similarity (ANOSIM) results for overall fecal microbiota compositions between control and experimental groups at the taxonomic level of genus.

Day −1	R	*p*-Value
PBS vs. Raw	0.025	0.311
PBS vs. Pasteurized	0.160	0.057
PBS vs. Skimmed	0.073	0.157
PBS vs. Pasteurized + ALP	−0.098	0.897
Raw vs. Pasteurized	0.286	**0.003**
Raw vs. Skimmed	0.288	**0.010**
Raw vs. Pasteurized + ALP	0.098	0.114
Pasteurized vs. Skimmed	−0.075	0.854
Pasteurized vs. Pasteurized + ALP	0.083	0.151
Skimmed vs. Pasteurized + ALP	−0.018	0.541
**Day 31**		
PBS vs. OVA	0.389	**0.002**
PBS vs. Raw	0.365	**0.005**
PBS vs. Pasteurized	0.388	**0.004**
PBS vs. Skimmed	0.610	**0.002**
PBS vs. Pasteurized + ALP	0.453	**0.004**
OVA vs. Raw	0.373	**0.003**
OVA vs. Pasteurized	0.174	**0.026**
OVA vs. Skimmed	0.464	**0.003**
OVA vs. Pasteurized + ALP	0.313	**0.005**
Raw vs. Pasteurized	0.266	**0.013**
Raw vs. Skimmed	0.122	0.106
Raw vs. Pasteurized + ALP	0.159	**0.049**
Pasteurized vs. Skimmed	0.326	**0.008**
Pasteurized vs. Pasteurized + ALP	0.214	**0.038**
Skimmed vs. Pasteurized + ALP	0.227	**0.034**

Global *R* comparison was based on ANOSIM performed within the software package Primer7 (PRIMER-E Ltd., Lutton, UK); *p*-values were calculated based on a permutational analysis vs. employing 999 permutations; square-root transformed. *p*-values < 0.05 are considered significant. PBS, phosphate-buffered saline; OVA, ovalbumin; raw, raw cow’s milk; pasteurized, pasteurized cow’s milk; skimmed, skimmed raw cow’s milk; pasteurized + ALP, pasteurized milk spiked with alkaline phosphatase.

**Table 2 ijms-22-03417-t002:** Significant average total count and percent relative abundance of sequences of individual fecal taxa between the different groups at day 31.

Taxonomic Levels	PBS ^1^ Count (% RA)	OVA ^2^ Count (% RA)	Raw ^3^ Count (% RA)	Pasteurized ^4^ Count (% RA)	Skimmed ^5^ Count (% RA)	Pasteurized + ALP ^6^ Count (% RA)	*p*-Value *	FDR *p*-Value *	Dunn’s Post Hoc Tests ^#^
**Phylum**									
Proteobacteria	312.17 (0.76)	575.13 (1.33)	179.88 (0.44)	910.38 (2.06)	244.00 (0.60)	811.50 (1.93)	**0.000**	**0.008**	**1 v 4, 1 v 6, 2 v 4, 2 v 6, 3 v 4, 3 v 6, 4 v 5, 5 v 6**
**Family**									
Burkholderiaceae	280.33 (0.68)	170.63 (0.37)	55.13 (0.14)	786.25 (1.78)	132.13 (0.32)	657.63 (1.56)	**0.000**	**0.005**	**1 v 4, 2 v 4, 2 v 6, 3 v 4, 3 v 6, 4 v 5, 5 v 6**
**Genus**									
*Blautia*	22.00 (0.08)	160.38 (0.37)	216.00 (0.54)	388.88 (0.87)	99.25 (0.23)	121.13 (0.29)	**0.003**	*0.066*	**1 v 2, 1 v 3, 1 v 4, 1 v 6, 3 v 5**
*Lachnospiraceae* NK4A136 group	1608.33 (5.87)	4399.75 (9.77)	6321.13 (15.59)	6045.00 (13.78)	9674.00 (24.60)	5868.88 (14.00)	**0.001**	**0.038**	**1 v 3, 1 v 4, 1 v 5, 1 v 6, 2 v 5**
*Lachnospiraceae* UCG-001	1175.67 (4.40)	796.13 (1.80)	2478.63 (6.10)	787.38 (1.79)	2000.75 (5.06)	2200.00 (5.40)	**0.003**	*0.066*	**2 v 3, 2 v 5, 2 v 6, 3 v 4, 4 v 5, 4 v 6**
*Lachnospiraceae* UCG-008	31.50 (0.13)	95.38 (0.21)	162.38 (0.40)	110.25 (0.25)	172.75 (0.42)	157.75 (0.38)	**0.002**	*0.054*	**1 v 2, 1 v 3, 1 v 4, 1 v 5, 1 v 6, 2 v 3**
*Oscillibacter*	42.00 (0.15)	144.88 (0.32)	179.63 (0.43)	220.63 (0.50)	308.50 (0.78)	130.63 (0.32)	**0.003**	*0.070*	**1 v 2, 1 v 3, 1 v 4, 1 v 5, 5 v 6**
*Parasutterella*	273.67 (0.66)	165.63 (0.36)	54.00 (0.14)	766.00 (1.73)	127.75 (0.31)	637.88 (1.51)	**0.000**	**0.004**	**1 v 3, 2 v 4, 2 v 6, 3 v 4, 3 v 6, 4 v 5, 5 v 6**
*Ruminiclostridium* 5	34.00 (0.13)	51.13 (0.12)	116.88 (0.28)	96.88 (0.22)	164.25 (0.41)	83.38 (0.20)	**0.002**	*0.054*	**1 v 3, 1 v 4, 1 v 5, 2 v 3, 2 v 4, 2 v 5**

Individual taxa differences between groups were assessed for significance using Kruskal–Wallis test on centered log ratio transformed (CLR) data, with false discovery rate (FDR)-corrected *p*-values reported. Adjusted *p*-values were considered significant at: * FDR-*p* ˂ 0.05; trends at * *p* ˂ 0.05 are reported. Dunn’s multiple group comparison’s post hoc tests significance indicated ^#^
*p* ˂ 0.05. ^1^ PBS, phosphate-buffered saline; ^2^ OVA, ovalbumin; ^3^ raw, raw cow’s milk; ^4^ pasteurized, pasteurized cow’s milk; ^5^ skimmed, skimmed raw cow’s milk; ^6^ pasteurized + ALP, pasteurized milk spiked with alkaline phosphatase; count = mean number of sequences in defined group; % RA, percent relative abundance in defined group; NS, non-significant.

## Data Availability

All data generated or analyzed during this study are included in this published article (and its Appendix A). Additional datasets used and/or analyzed during the current study are available from the corresponding author on reasonable request. Raw sequence data are deposited in the National Center for Biotechnology Information (NCBI) Sequence Read Archive under the BioProject identifier PRJNA576858.

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
