# Peer review of "Raw Milk-Induced Protection against Food Allergic Symptoms in Mice Is Accompanied by Shifts in Microbial Community Structure"

_ijms, 2021, doi:10.3390/ijms22073417_

Round 1

Reviewer 1 Report

The authors present data on microbiome analyses of fecal pellets collected in previous murine models of food allergy. In these previous studies, mice were fed cow’s milk after different processing steps: no processing, only skimming, pasteurization and pasteurization + addition of alkaline phosphatase. Raw milk treatment prevented from experimental food allergy and pasteurization with alkaline phospatase partially prevented from food allergy. Fecal pellets were analyzed by 16S rRNA sequencing and gut microbiome composition was compared between different treatment groups before and after treatment. Treatment with raw milk was associated with an increased relative abundance of Clostridiales and a decreased relative abundance of Proteobacteria

Comments:

  1. The authors often refer to their previous publication (reference 20) and state that full protection from experimental food allergy was achieved by reconstituting pasteurized cow’s milk with alkaline phosphatase. This is not correct, as treatment with pasteurized cow’s milk + alkaline phosphatase was associated with a decrease in ear swelling but not a decrease in OVA-specific IgE, a hallmark of food allergy. This treatment also resulted in an increase of MNL- Th2 cells. Therefore, the authors should write about a partial, but not fully preventive effect.
  2. The manuscript is in part confusing because of multiple comparisons between groups. The authors should focus on comparing the group of raw milk treatment with the pasteurized treatment group with the untreated OVA and PBS groups, because these treatments were effective or ineffective in preventing from experimental food allergy in previous work. Both other groups (raw + skimmed and pasteurized + alkaline phosphatase gave mixed results). Such approach would moreover reduce multiple testing.
  3. It is interesting that experimental food allergy in itself (in the OVA group) without any milk treatment induced changes in the gut microbiome. These changes should be highlighted first before discussing the effects of the two treatment arms. Itisi unfortunate that no baseline information (day-9) is given for the OVA group, so it is unclear whether high richness is due to the allergenisation process or was present at baseline. This limitation should be acknowledged in the discussion section.
  4. While I understand that the wish is to identify single bacteria that might be supplemented as probiotics, microbiome studies have highlighted that clusters of bacteria do not come in isolation. Thus cluster analyses such as eg by Dirichtlet or other clustering methods should be undertaken in cross-sectional as well as longitudinal studies. Such clustering would also help to see whether there are clear differences in the gut microbial composition between groups.
  5. The milk was produced on a dairy farm in Ireland, but processed at Danone in Utrecht if I understand correctly. How was the milk shipped from Ireland to Utrecht ? Do shipment conditions impact results ?
  6. When discussing potential translation to humans a recent publication in Nature Medicien by Depner and colleagues should be referenced.
  7. Please state in lines 135-137 the direction of the effect, namely that richness was higher in pasteurized samples as compared to all others.
  8. Please spell out CT in legend of figure 1.
  9. Please include rarefaction curves in the suppl material.
  1.  

Author Response

The authors would like to thank reviewer 1 for the valuable comments on our manuscript. We addressed the questions raised by reviewer 1 by answering the specific comments point-by-point (please see attachment).

Reviewer 2 Report

This is an interesting study that goes beyond the simple raw milk vs pasteurized milk by adding ALP.

The authors discuss the primary concern, that if people consume unpasteurized milk, they may be exposed to pathogenic bacteria and become ill/infected.

An important thing to consider is repeatability withe different collections of unpasteurized milk. Is it possible that the single collection from one farm in Ireland had a unique microbiota that has this effect and only in mice?  I hope that you would consider a broader study to look at different farms, and then to move into the human experience if one can find a way to adequately protect human consumers from pathology.

I suggest adding a short warning at the conclusion that states a need to perform additional tests in the future with different raw cows milk.  It is also useful to consider measuring the microbial population in the cow's milk and across farms that produce raw cows milk.

Author Response

The authors would like to thank reviewer 2 for the valuable comments on our manuscript. We addressed the questions raised by reviewer 2 by answering the specific comments point-by-point (please see attachment).
